# Multisystem Immune-Related Adverse Events from Dual-Agent Immunotherapy Use

Yuchen Li [1], Gregory Pond [2] and Elaine McWhirter [2,*]

1 Department of Oncology, McMaster University, Hamilton, ON L8V 5C2, Canada; yuchen.li@medportal.ca
2 Ontario Institute for Cancer Research, Toronto, ON M5G 0A3, Canada; gpond@mcmaster.ca
* Correspondence: emcwhirt@hhsc.ca

**Abstract:** Background: little is known about the incidence and characteristics of multisystem immune-related adverse events (irAEs) associated with dual-agent ipilimumab and nivolumab use. Methods: A retrospective cohort review was completed that included cancer patients seen at the Juravinski Cancer Centre who received at least one dose of ipilimumab and nivolumab from 2018 to 2022. Patient characteristics, cancer types, and irAEs were recorded. Multivariate logistic and cox regressions were completed, comparing those who developed multisystem irAEs, single irAE, and no irAE. Results: A total of 123 patients were included in this study. Out of 123 patients, 72 (59%) had melanoma, 50/123 (41%) had renal cell carcinoma (RCC), and 1/123 (1%) had breast cancer. Multisystem irAEs were seen in 40% of the overall cohort. The most common irAE type was dermatitis (22%), followed by colitis (19%) and hepatitis (17%). Conclusions: Our study demonstrated that multisystem irAEs are prevalent amongst patients receiving ipilimumab and nivolumab. It is important for both physician education and the counseling and consent of patients to monitor the potential for multiple irAEs.

**Keywords:** immunotherapy; immune-related adverse events; ipilimumab and nivolumab; melanoma; renal cell carcinoma





## 1. Introduction

Programmed cell death ligand 1 (PD-L1)/program cell death protein 1 (PD-1) and cytotoxic lymphocyte-associated antigen 4 (CTLA-4) inhibitors are two of the most common immunotherapy classes used in cancer therapeutics [1]. In clinical trials, the use of immunotherapy for specific cancer types has shown improved outcomes when combined with or in place of chemotherapy [2,3]. More recently, it has been observed that combining PD-(L)1 and CTLA-4 inhibitors can further improve outcomes in certain cancers over single-agent immunotherapy or standard of care treatment alone. CHECKMATE 067 was a landmark trial that established the use of ipilimumab plus nivolumab for suitable patients with metastatic melanoma. In this study, the combination was compared to either nivolumab or ipilimumab as single agents, and we found a significant improvement in 5-year overall survival in favour of the combination arm [4]. For patients with intermediate or poor-risk untreated advanced clear cell RCC, combination immunotherapy also demonstrated superior OS and PFS compared to standard treatment with sunitinib in the CHECKMATE 214 study [5]. The positive results from these landmark clinical trials cemented the use of ipilimumab plus nivolumab as a first-line treatment for various cancer types in the metastatic setting [4–8]. Though effective, dual-agent immunotherapy often results in significant immune-related adverse events (irAEs). Clinical trials involving combination ipilimumab and nivolumab for treatment of metastatic melanoma and RCC have reported rates of 45–59% for grade 3 or higher irAEs during the course of treatment [4,5].

In addition to high rates of irAEs, it has been observed clinically that multiple different irAEs can occur during or/and after treatment, a phenomenon described as multisystem irAEs. Two studies that examined multisystem irAEs in patients treated with single-agent

immunotherapy reported rates ranging from 5 to 17% [9,10]. Furthermore, it appears that the occurrence of multisystem irAEs is associated with improved survival outcomes compared to those who did not experience irAEs [9].

There is currently a paucity of research on the incidence of multisystem irAEs for patients treated with dual-agent immunotherapy; importantly, these are not reported in the landmark clinical trials [4,5]. In this study, we set out to assess the rate, type, and timing of multisystem irAEs in patients with solid tumours treated with ipilimumab and nivolumab at our institution. Additionally, we compared response rate, progression-free survival (PFS), and overall survival (OS) outcomes between patients who experienced multisystem irAEs and those who did not. We also endeavored to identify any predictive laboratory or clinical variables for the development of multisystem irAEs.

## 2. Materials and Methods

### 2.1. Patient Population

A retrospective cohort study was completed that included all adult cancer patients (age 18 or older) seen at the Juravinski Cancer Centre who received at least one dose of ipilimumab and nivolumab from 1 January 2018 to 31 May 2022. Baseline characteristic information, including sociodemographic (age, sex, and Eastern Cooperative Oncology Group (ECOG) performance), cancer type and staging, treatment-related, and outcome information were recorded.

### 2.2. Treatment

Ipilimumab and nivolumab dose regimens that patients received are the standard doses approved in Canada. For the melanoma cohort, ipilimumab was given at 3 mg per kilogram (kg) intravenously (IV) and nivolumab at 1 mg per kg IV every 3 weeks for up to 4 doses, followed by maintenance nivolumab at 3 mg per kg IV every 2 weeks (or 6 mg per kg IV every 4 weeks) until disease progression or unacceptable toxicities occurred. For the renal cell carcinoma cohort, the dosing was ipilimumab at 1 mg per kg IV and nivolumab at 3 mg per kg IV every 3 weeks for up to 4 cycles, followed by maintenance nivolumab at 3 mg per kg IV every 2 weeks (or 6 mg per kg IV every 4 weeks) until disease progression or unacceptable toxicities occurred.

### 2.3. Outcomes

We defined multisystem irAEs as having 2 or more irAEs affecting different organ systems during the course of immunotherapy treatment, either during the combination phase with ipilimumab and nivolumab or during the maintenance phase with nivolumab alone. We included all irAEs that presented sequentially, with no overlapping period between onset and resolution of an irAE, or/and irAEs that presented concurrently, when there is at least 1 day of overlap between two or more irAEs.

### 2.4. Statistical Analysis

Statistical analysis was performed using SAS v9.2 and R v4.2.0. Logistic regression analyses were completed to assess for any clinical or laboratory factors that predict the development of multisystem irAEs. Progression-free survival (PFS) and overall survival (OS) estimates were estimated using the Kaplan–Meier method. A 180-day landmark analysis was performed to explore the relationship between patients who experienced multisystem irAEs, a single irAE, and no irAE with PFS and OS.

## 3. Results

### 3.1. Baseline Characteristics

A total of 123 patients were included in this study. The baseline characteristics are shown in Table 1. The mean age of diagnosis was 57.6 (SD = 11.4) years old, with the majority being male (69%). Most patients had good performance status (ECOG 0–1) and stage 4 disease prior to treatment start. Three patients with stage III melanoma were deemed to be unresectable

stage III and received ipilimumab and nivolumab as palliative treatment. A total of 72 of out 123 (59%) patients had melanoma, 50/123 (41%) had RCC, and 1 patient had metastatic breast cancer and received ipilimumab and nivolumab as part of a clinical trial.

**Table 1.** Characteristics of patients at baseline.

| | | All Patients | Melanoma | RCC |
|---|---|---|---|---|
| N | | 123 | 72 | 50 |
| Age—year | Mean (sd) | 57.6 (11.4) | 55.7 (12.2) | 60.6 (9.6) |
| | Median (IQR) | 59 (25, 83) | 57.5 (25, 83) | 63 (36, 77) |
| Sex—N (%) | Male | 85 (69.1) | 47 (65.3) | 37 (74.0) |
| | Female | 38 (30.9) | 25 (34.7) | 13 (26.0) |
| ECOG—N (%) | 0 | 66 (60.6) | 39 (66.1) | 26 (53.1) |
| | 1 | 37 (33.9) | 19 (32.2) | 18 (36.7) |
| | 2 | 5 (4.6) | 0 | 5 (10.2) |
| | 3 | 1 (0.9) | 1 (1.7) | 0 |
| BRAF *—N (%) | V600E | | 22 (30.6) | |
| | V600K | | 7 (9.7) | |
| | Other BRAF mutation | | 5 (6.9) | |
| | None | | 38 (52.8) | |
| Stage—N (%) | 1 | 0 | 0 | 0 |
| | 2 | 0 | 0 | 0 |
| | 3 | 3 (2.4) | 3 (4.2) | 0 |
| | 4 | 120 (97.6) | 69 (95.8) | 50 (100) |

* BRAF = v-raf murine sarcoma viral oncogene homolog B1.

## 3.2. Treatment-Related

### 3.2.1. Treatment Received

All patients completed at least one cycle of ipilimumab and nivolumab, with 45% of patients receiving all four cycles of combination therapy. A total of 57% of patients received subsequent nivolumab, with a median of eight cycles of maintenance treatment (Table 2).

**Table 2.** Immunotherapy treatment and immune-related adverse events.

| | | All Patients | Melanoma | RCC |
|---|---|---|---|---|
| N | | 123 | 72 | 50 |
| | Median (IQR) | 3 (2, 4) | 2 (2, 4) | 4 (3, 4) |
| Number of cycles of combination ipilimumab and nivolumab received—N (%) | 1 | 17 (15.0) | 11 (17.5) | 6 (12.0) |
| | 2 | 27 (23.9) | 21 (33.3) | 6 (12.0) |
| | 3 | 18 (15.9) | 10 (15.9) | 8 (16.0) |
| | 4 | 51 (45.1) | 21 (33.3) | 30 (60.0) |
| Received at least one cycle of maintenance nivolumab—N (%) | Yes | 65 (57.0) | 34 (53.1) | 31 (62.0) |
| Number of irAEs experienced per patient—N (%) | 0 | 34 (27.6) | 15 (20.8) | 18 (36.0) |
| | 1 | 40 (32.5) | 23 (31.9) | 17 (34.0) |
| | 2 | 33 (26.8) | 22 (30.6) | 11 (22.0) |
| | 3 | 14 (11.4) | 10 (13.9) | 4 (8.0) |
| | 4 | 2 (1.6) | 2 (2.8) | 0 |
| Time to onset of first irAE—days | Median (IQR) | 41 (19, 80) | 35 (16, 53) | 54 (22, 110) |
| | Range | 1, 460 | 5, 367 | 1, 460 |
| Concurrent or/and sequential irAE—N of patient (%) | Sequential | 20 (40.8) | 14 (41.2) | 6 (40.0) |
| | Concurrent | 18 (36.7) | 12 (35.3) | 6 (40.0) |
| | Both sequential and concurrent | 11 (22.4) | 8 (23.5) | 3 (20.0) |
| Types of irAE—N (%) | Adrenal | 3 | 3 | 0 |
| | Colitis | 29 | 19 | 10 |
| | Cutaneous | 34 | 21 | 13 |
| | Hepatotoxicity | 26 | 21 | 5 |
| | Hyperthyroidism | 7 | 4 | 3 |
| | Hypothyroidism | 16 | 9 | 7 |
| | Hypophysitis | 3 | 1 | 2 |
| | Neurological | 6 | 5 | 1 |
| | Pneumonitis | 9 | 5 | 4 |
| | Renal | 6 | 6 | 0 |
| | Rheumatological | 12 | 7 | 5 |
| | Ocular | 1 | 1 | 0 |
| | Pancreatitis | 2 | 1 | 1 |
| | Hematological | 1 | 1 | 0 |
| | Myocarditis | 1 | 1 | 0 |
| Systemic steroid use—N (%) | First irAE | 64/89 (71.9) | 40/57 (70.2) | 24/32 (75.0) |
| | Second irAE | 39/48 (81.3) | 27/33 (81.8) | 12/15 (80.0) |
| | Third IrAE | 14/16 (87.5) | 11/12 (91.7) | 3/4 (75.0) |
| | Fourth irAE | 1/2 (50.0) | 1/2 (50.0) | 0/0 (0.0) |

### 3.2.2. Immune-Related Adverse Events

Seventy-two percent of patients had at least one irAE during or after treatment, with a higher rate seen in the melanoma cohort (79% of patients) than in the RCC cohort (64% of patients). Multisystem irAEs were seen in 40% of the overall cohort (47% of the melanoma cohort; 30% of the RCC cohort), with most patients having two irAEs (Table 2). The median time to onset of the first irAE was 54 days for the RCC cohort and 35 days for the melanoma cohort. Most irAEs occurred during the combination treatment phase with ipilimumab and nivolumab, specifically, in 89%, 59%, 75%, and 50% of cases of first, second, third, and fourth irAEs, respectively. Of the 49 out of the 123 patients who experienced multisystem irAEs, 20/49 (41%) patients experienced irAEs sequentially, with no periods of overlap. Conversely, 18/49 (37%) of the patients experienced only concurrent irAEs, while 11/49 (22%) of the patients had both concurrent and sequential irAEs.

The most common types of irAEs were dermatitis, colitis, hepatitis, and thyroiditis. Patients who experienced multisystem irAEs had less rates of ir-hepatitis (14% vs. 23%) and dermatitis (20% vs. 26%), but more occurrence of rare irAEs such as pancreatitis (2 vs. 0%), myocarditis (1% vs. 0%), and hypophysitis (2.5% vs. 0%); see Figure 1. The majority of patients, 76%, required systemic steroid therapy for the treatment of irAE(s); see Table 2. A total of 89 out of 123 (72%) patients treated with ipilimumab and nivolumab required hospitalization. Just over a third (34%) of these hospitalizations were confirmed to be due to irAEs.

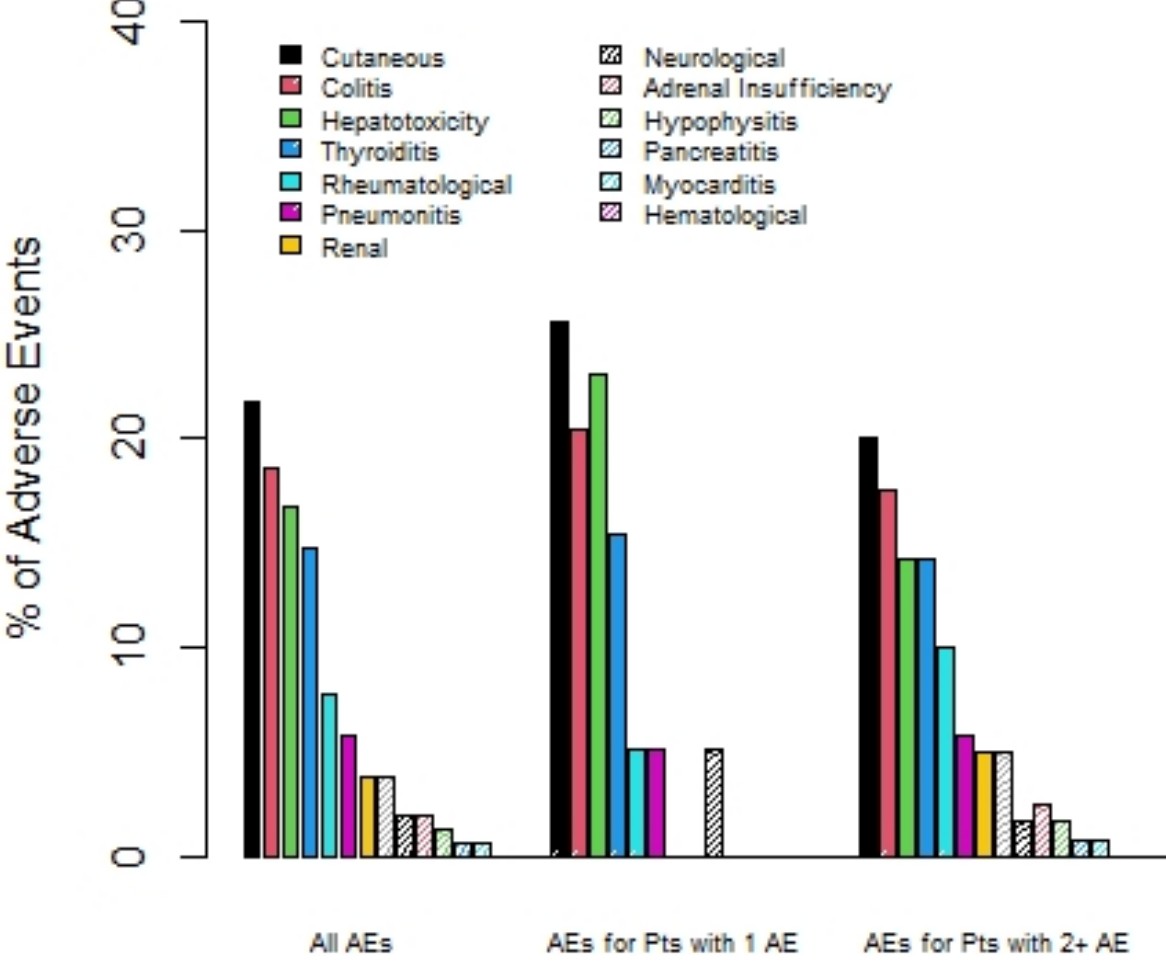

**Figure 1.** Percentage of irAEs by type, comparing overall group, patients with 1 irAE only, and those with 2 or more irAEs.

### 3.2.3. Response Rate

Using a landmark analysis of 18 days, 16/41 (39.0%), 29/42 (69.1%), and 32/40 (80.0%) patients had clinical benefit amongst those with zero, one, or two plus irAEs, respectively.

### 3.2.4. Progression-Free and Overall Survival

Of the 72 melanoma patients, 43 were eligible for the landmark analysis for OS. They had a median (95% CI) OS of 35.3 (13.9 to not reached (NR)) months. The median (95% CI) PFS was 35.3 (18.8 to NR) months, from the day 180 landmark date, for the 37 melanoma patients eligible for the PFS analysis. One-year (95% CI) OS was 60.0% (19.6% to 85.2%), 92.3% (56.6% to 98.9%), and 83.1% (56.1 to 94.3%) for patients with zero, one, or two plus irAEs prior to day 180 (see Figure 2). There was no statistically significant difference in OS among those with zero, one, or two irAEs (*p* = 0.23). The results were similar based on PFS (Figure 3), with no statistically significant difference among those with zero, one, or two irAEs (*p* = 0.20).

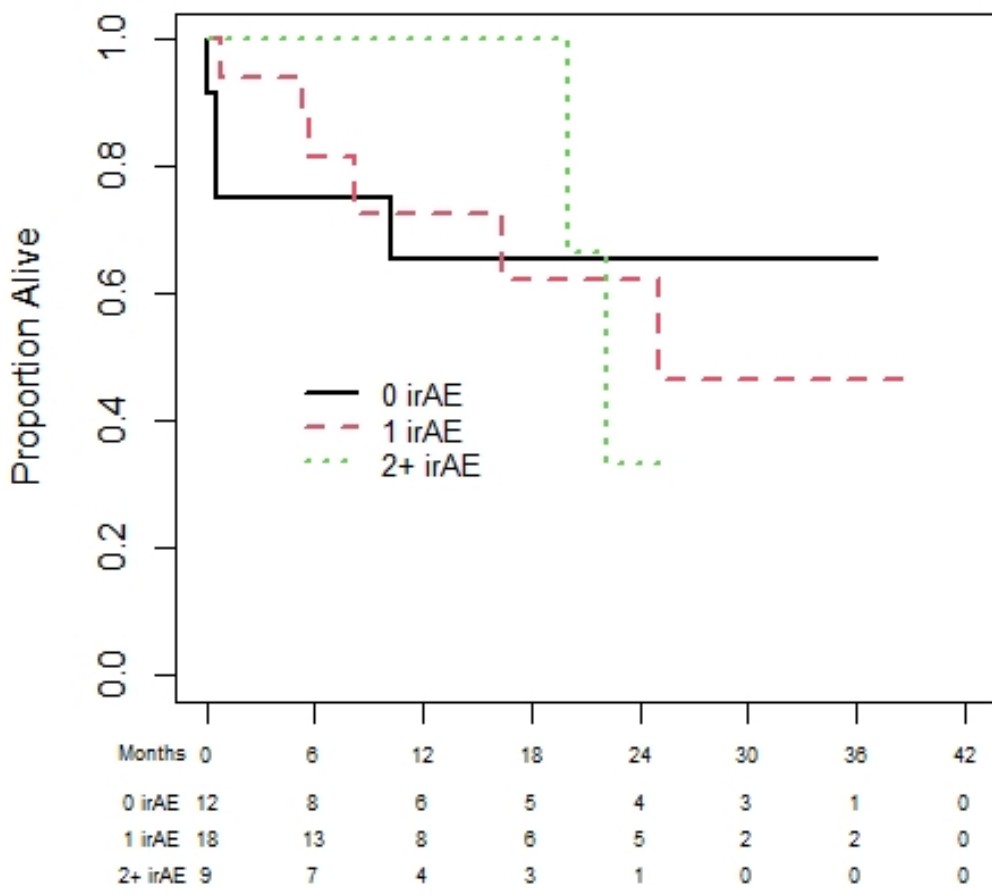

**Figure 2.** Overall survival of melanoma cohort based on 180-day landmark analysis.

Amongst the 50 patients with RCC, 39 and 27 patients were eligible for the OS and PFS landmark analyses, respectively. The median OS was NR and median (95% CI) PFS was 25.1 (17.4 to NR) months. The 1-year (95% CI) OS for patients with zero, one, and two plus irAEs was 65.6% (32.0% to 85.6%), 72.5% (41.1% to 89.0%), and 100% (100% to 100%), with no statistically significant difference in OS among those with zero, one, or two irAEs (*p* = 0.87); see Figure 4. Similarly, the results for PFS (see Figure 5) show no significant difference in PFS among those with zero, one, or two irAEs (*p* = 0.65).

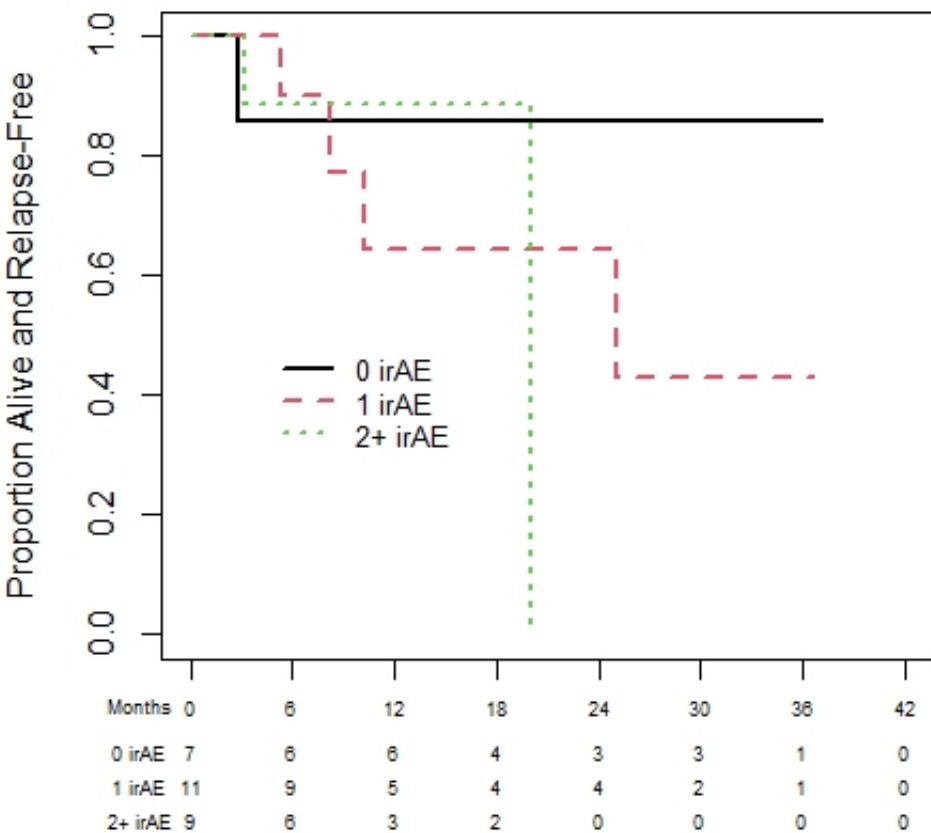

**Figure 3.** Progression-free survival of melanoma cohort based on 180-day landmark analysis.

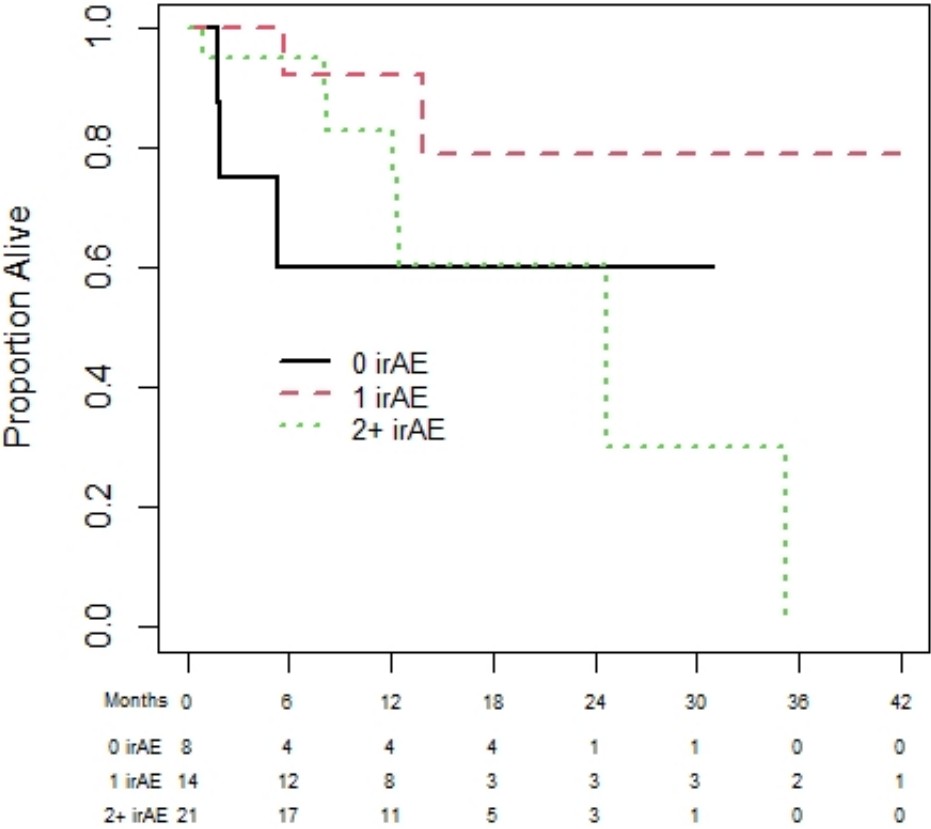

**Figure 4.** Overall survival of renal cell carcinoma cohort based on 180-day landmark analysis.

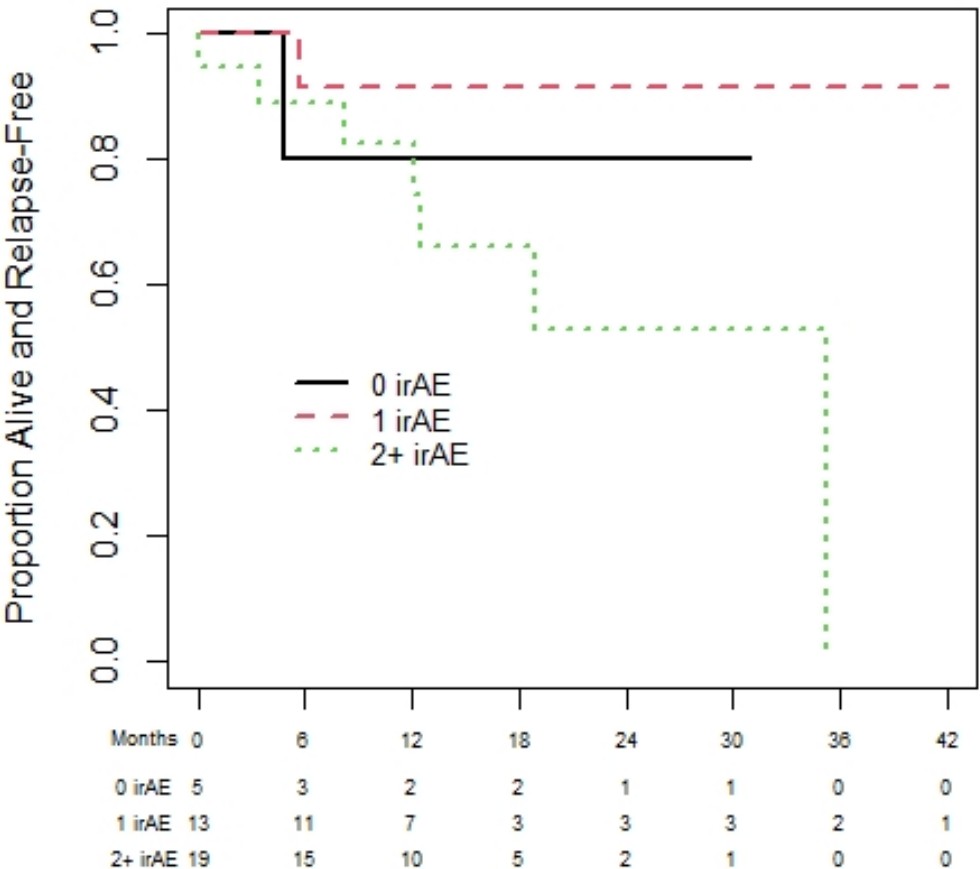

**Figure 5.** Progression-free survival of renal cell carcinoma cohort based on 180-day landmark analysis.

*3.3. Predictor(s) of Having Multisystem irAEs*

Multiple variables were assessed, including age, sex, cancer type, baseline laboratory markers (neutrophil or lymphocyte count; LDH), and BRAF mutation status. There appeared to be no statistically significant clinical or laboratory predictor for the development of multisystem irAEs (Table 3).

**Table 3.** Logistic regression comparing patients who experienced ≥2 irAEs versus those who did not.

|  |  | Odds Ratio (95% CI) | *p*-Value |
|---|---|---|---|
| Age | year | 1.01 (0.97, 1.04) | 0.73 |
| Sex | M vs. F | 0.87 (0.40, 1.89) | 0.73 |
| Cancer Type | Melanoma vs. RCC | 2.09 (0.98, 4.47) | 0.058 |
| Prior Surgery | Y vs. N | 0.93 (0.44, 1.94) | 0.85 |
| Overall Stage | 1 | 3.71 (0.68, 20.35) | 0.19 |
|  | 2 | 1.48 (0.44, 5.03) |  |
|  | 3 | 0.52 (0.19, 1.48) |  |
|  | 4 | Reference |  |
| Neutrophils | unit | 0.96 (0.86, 1.08) | 0.50 |
| Lymphocytes | unit | 0.98 (0.76, 1.26) | 0.87 |
| NLR (neutrophil-to-lymphocyte ratio) | log-unit | 0.72 (0.42, 1.23) | 0.23 |
| LDH | unit | 1.00 (1.00, 1.00) | 0.93 |
| BRAF | N vs. Y | 1.20 (0.46, 3.09) | 0.71 |

**4. Discussion**

Dual-agent immunotherapy with ipilimumab and nivolumab is now the preferred initial treatment regimen for many patients with metastatic melanoma, RCC, and NSCLC [3,6,7]. Its usage is expected to continue to rise in the future [4–6]. To our knowledge, this study is the first to examine the rate and characteristics of multisystem irAEs associated with combined ipilimumab and nivolumab use. We found that 40% of patients experienced

multisystem irAEs during their course of treatment. This rate is much higher than the rate of multisystem irAEs reported with single-agent immunotherapy by Shanker et al. [9]. In their multicentre cohort study of patients with advanced NSCLC, the majority of patients received single-agent PD-(L)1 inhibitor, with only 9% of patients developing multisystem irAEs.

Our study corroborates prior findings that CTLA-4 inhibition leads to higher rates of irAEs than PD-(L)1 inhibition, potentially in a dose-dependent manner [11]. In our study, patients in the melanoma cohort received a higher dose of ipilimumab (3 mg per kg) than those in the RCC cohort (1 mg per kg). Notably, the rate of multisystem irAEs was 17% higher in the melanoma cohort compared to the RCC cohort. CHECKMATE 067, which used 3 mg per kg dosing of ipilimumab, reported rates of grade 3 or greater irAEs of 59%, compared with CHECKMATE 214, which used 1 mg per kg of ipilimumab and reported rates of 46% [4,5]. It has been reported that CTLA-4 inhibitors block earlier stages of T cell activation in comparison to PD-(L)1 inhibitors, affecting a broader range of T cells and leading to more widespread and more intense immune reactions [12]. The mechanism behind the greater incidence of multisystem irAEs with combination immunotherapy remains to be elucidated. However, our understanding of the pathophysiology of immunotherapy allows us to postulate some reasons for higher occurrences of multisystem irAEs in this group. Dual blockade of CTLA-4 and PD-(L)1 leading to more potent activation of T cells may result in higher rates of irAEs that impact different organs systems simultaneously. In addition, CTLA-4 inhibitors, such as ipilimumab, and PD-(L)1 inhibitors, such as nivolumab, result in irAEs preferentially in certain organ systems. For example, CTLA-4 inhibitors are associated with higher rates of colitis and hypophysitis, whereas PD-(L)1 inhibitors more frequently lead to thyroiditis, pneumonitis, and autoimmune diabetes [13]. When these therapies are combined, there may be a broader spectrum of irAEs, resulting in an increased incidence of multisystem manifestations. Lastly, it may be that multisystem irAEs all derive from a shared common pathobiology, including a common human leukocyte antigen (HLA) gene or autoantibody formation [9].

We found that a majority of multisystem irAEs occur during the combination phase of treatment with ipilimumab and nivolumab. The median time to onset of the first irAE was 41 days, comparable to the 49 days reported by Shankar et al. for single-agent immunotherapy. Furthermore, we identified that multisystem irAEs from combination immunotherapy can occur either sequentially, concurrently, or a combination of both. In contrast, with PD-(L)1 inhibition alone, Shankar et al. found that multisystem irAEs presented sequentially, with no overlapping period of concurrent toxicities [9].

The most common irAEs that we identified for patients experiencing multisystem irAEs were dermatitis, colitis, hepatotoxicity, and thyroiditis, similar to the findings for multisystem irAEs with single-agent PD-(L)1 inhibitors [9]. However, in comparison to the types of irAE for patients treated with single-agent PD-(L)1 inhibitors, our cohort had lower rates of pneumonitis and a higher rate of colitis. This could be attributed to the addition of the CTLA-4 inhibitor, as it is more frequently linked to colitis and less so to pneumonitis [13]. In addition, we found a higher occurrence of rare irAEs, such as hypophysitis, pancreatitis, and myocarditis, when they occurred as part of multisystem irAE phenomena.

We undertook a 180-day landmark analysis to examine the outcomes of patients who experienced multisystem irAEs in comparison to those who did not. The use of a landmark analysis is important to eliminate survivorship bias [14]; patients who have a better response are likely to remain on immunotherapy longer, which in turn creates an increased risk of irAEs. Using this analysis, we found no significant difference in the PFS or OS for those who experienced multisystem irAEs and those who did not. In contrast, Shankar et al. reported that patients treated with single-agent PD-(L1) inhibitors who experienced multisystem irAEs showed better OS (HR 0.57) and PFS (HR 0.67) than those with no irAE [9], though a landmark analysis was not performed. Similarly, a retrospective cohort study by Serna-Higuita et al. found that patients with melanoma

treated with ipilimumab and nivolumab who experienced irAEs had better disease control and prolonged OS [12]. While the exact mechanism is still not well understood, it has been postulated that the processes causing irAEs are the same as those employed to kill cancer cells. Consequently, the occurrence of multisystem irAEs may indicate that the immune system is working effectively to eliminate the underlying cancer as well [15].

Importantly, our study suggested that irAEs can introduce significant comorbidities for patients, and over three-quarters required multiple rounds of systemic corticosteroid treatments to address multiple irAEs. Long-term use of steroids can lead to side effects including osteoporosis, cardiovascular disease, and metabolic disturbances [16]. Additionally, treatments, including the use of steroids, incur significant costs for both patients and the healthcare system. It is estimated that, excluding admission-related expenses, the cost per patient for treating irAEs associated with dual-agent immunotherapy is approximately CAD 400 [16]. Immune-related adverse events can be quite severe; we found that over one-third of patients in our study required hospitalization relating to irAEs. Balaji et al. reported that around 20% of patients treated with immunotherapy require inpatient hospitalization, with most of these attributed to irAEs. The hospitalization rates for patients receiving dual-agent immunotherapy treatment were notably higher, estimated to have more than a six-fold increase in the odds of hospital admission compared to those treated with single-agent immunotherapy [17,18]. As reported by a single institute study from Massachusetts General Hospital in the United States, the annual cumulative cost of admissions related to irAEs easily reaches into the millions, exceeding USD 3000 per day of admission [19]. The cost can often reach upwards of USD 20,000 per irAE, as found in a study of elderly patients with melanoma; the highest costs were associated with respiratory, central nervous system (CNS), and endocrinopathy-related irAEs [20].

The identification of reliable clinical or laboratory predictors for the development of multisystem irAEs in patients receiving ipilimumab and nivolumab remains elusive. A few smaller studies have suggested that male sex, absolute lymphocyte account, and neutrophil-to-lymphocyte ratio may predict higher incidences of irAEs [21,22]. However, our analysis did not support these findings.

Our study has several important limitations. It is a chart review from a single academic cancer centre and is biased by its retrospective nature. We did not collect data on lifestyle factors, such as smoking and alcohol consumption, or concomitant medications, all of which could potentially impact the development of irAEs. We categorized and graded irAEs using the Common Terminology Criteria for Adverse Events (CTCAE), referencing clinical notes, laboratory findings, and imaging results. However, we recognize that the characterization and grading may be subjective, particularly for irAEs without laboratory parameters, such as pruritus and joint pain. Importantly, despite the extensive data collection for irAEs in the published, large, randomized phase III trials of combination immunotherapy, the incidence of multisystem irAEs was not reported. This underscores a significant gap in the literature, which could be remedied by analysis and publication of previously collected clinical trial data, and standardization of irAE reporting for future trials.

## 5. Conclusions

In summary, to our knowledge, this is the first report in the literature documenting the rates and characteristics of multisystem irAEs in patients receiving combination PD-1 and CTLA-4 immunotherapy. Given the high rates, early onset, and presentations of concurrent and/or sequential multisystem irAEs, our study highlights the need for clinicians to have more extensive risk–benefit discussions with patients prior to starting dual-agent immunotherapy. Both patients and clinicians need to be educated on the identification of multisystem irAEs to ensure timely intervention for these potentially debilitating toxicities. Additional research, such as identifying predictors for the onset of irAEs and the impact of enhanced monitoring strategies, is essential to mitigate the toxicity associated with irAEs. Future clinical trials should clearly report the incidence of multisystem irAEs and incorporate quality of life (QoL) metrics into their designs.

**Author Contributions:** Conceptualization, E.M. and Y.L.; methodology, E.M. and Y.L.; software, G.P.; formal analysis, G.P.; investigation, E.M.; data curation, Y.L.; writing—original draft preparation, Y.L.; writing—review and editing, Y.L., E.M. and G.P.; supervision, E.M. All authors have read and agreed to the published version of the manuscript.

**Funding:** This research received no external funding.

**Institutional Review Board Statement:** This study was conducted in accordance with the Declaration of Helsinki and approved by the Hamilton Integrated Research Ethics Board. Protocol code (HiREB 15011) on 21 June 2022.

**Informed Consent Statement:** This study was approved by the Hamilton Integrated Research Ethics Board (HiREB). Patient consent was waived as many patients are deceased, and it is impractical to obtain consent. The research ethics board permitted this.

**Data Availability Statement:** The original contributions presented in the study are included in the article, further inquiries can be directed to the corresponding author.

**Conflicts of Interest:** Y.L. and G.P. have no conflicts of interest to disclose. E.M.: advisory boards for BMS, Merck, Novartis, EMD Serrono, Pfizer, and Sanofi-Genzyme.

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
