# Peer review of "Multisystem Immune-Related Adverse Events from Dual-Agent Immunotherapy Use"

_curroncol, doi:10.3390/curroncol31010028_

Round 1
Reviewer 1 Report (Previous Reviewer 1)
Comments and Suggestions for Authors Authors have been answered and revised my most concerns. The revision of only question 4) 0,6,12,24----of horizontal ordinate in Figure 3,4,5,6 is months or weeks is unclear . The location of months should be added in the front of 0,6,12,24 .After authors revised question 4) , it can be accepted.
Author Response
Authors have been answered and revised my most concerns. The revision of only question 4) 0,6,12,24--of horizontal ordinate in Figure 3,4,5,6 is months or weeks is unclear . The location of months should be added in the front of 0,6,12,24. After authors revised question 4) , it can be accepted.
• Response: Thank-you for your suggestion. We have revised Figure 3,4,5,6 (now Figure 2,3,4,5); the x-axis are now clearly display the time in months. Additionally, we have repositioned the “months” label in front of 0,6,12,24 months mark to improve clarity
Reviewer 2 Report (Previous Reviewer 3)
Comments and Suggestions for Authors
This retrospective cohort study investigated the incidence and characteristics of multisystem immune-related adverse events (irAEs) in cancer patients who received a combination of ipilimumab and nivolumab from 2018 to 2022. The study included 123 patients, primarily with melanoma and renal cell carcinoma, and found that 40% of patients experienced multisystem irAEs, with dermatitis, colitis, and hepatitis being the most common types. The findings emphasize the importance of monitoring and educating both physicians and patients about the potential for multiple irAEs when using this combination therapy.
The author statement that “Little is known about the incidence and characteristics of multisystem immune10 related adverse events (irAEs) associated with dual agent ipilimumab and nivolumab use” in abstract. However, It’s well known that combination of ipilimumab and nivolumab can induce immune-related adverse events (DOI:10.1007/s11523-021-00825-2, https://doi.org/10.1007/s10637-022-01305-8 ). Can you explain the difference between other research? Such as : https://doi.org/10.2147/TCRM.S193338
The fig 1 is nor clear, it should be better to change it with high resolution.
The picture is cluttered and difficult to read. What’s difference between the two pictures in page 14 ?
It would be better separate the conclusion from discussion part.
Comments on the Quality of English LanguageModerate editing of English language required
Author Response
This retrospective cohort study investigated the incidence and characteristics of multisystem immunerelated adverse events (irAEs) in cancer patients who received a combination of ipilimumab and nivolumab from 2018 to 2022. The study included 123 patients, primarily with melanoma and renal cell carcinoma, and found that 40% of patients experienced multisystem irAEs, with dermatitis, colitis, and hepatitis being the most common types. The findings emphasize the importance of monitoring and educating both physicians and patients about the potential for multiple irAEs when using this combination therapy. The author statement that “Little is known about the incidence and characteristics of multisystem immune10 related adverse events (irAEs) associated with dual agent ipilimumab and nivolumab use” in abstract. However, It’s well known that combination of ipilimumab and nivolumab can induce immunerelated adverse events (DOI:10.1007/s11523-021-00825-2, https://doi.org/10.1007/s10637-022-01305-8 ). Can you explain the difference between other research? Such as: https://doi.org/10.2147/TCRM.S193338
• Response: It is correct that there are many studies that examined the overall combined rates of irAEs relating to ipilimumab and nivolumab use. However, to our knowledge, what has not been previously reported is the incidence of multisystem ieAEs. These are defined by the presence of two or more irAEs affecting different organ systems during immunotherapy treatment and occur concurrently, sequentially or a combination of both. For example, a patient may be receiving steroids for colitis, and during steroid taper for resolving colitis, then develop hepatitis or arthritis requiring re-escalation of steroids. Our study found a 40% incidence of multisystem irAEs, underscoring its importance for patient
education and consent.
The fig 1 is nor clear, it should be better to change it with high resolution.
• Response: Thank-you; we replaced the original figure with a higher resolution picture.
The picture is cluttered and difficult to read. What’s difference between the two pictures in page 14 ?
• Response: Are you referring to Figure 1? We replaced all the figures with higher-resolution versions. Please see caption for explanations for these figures.
It would be better separate the conclusion from discussion part.
• Response: Thank you for this comment. We have now included a new section entitled “Conclusions”, and separated the conclusion from discussion.
Round 2
Reviewer 1 Report (Previous Reviewer 1)
Comments and Suggestions for Authors
It has no obvious improved.
In table 2. The location of Median (IQR) is not suitable.The numbmers of patients are inconsistent among different items.
Figure 2-4, title , the number under table with the survival curve are in consistent.
Comments on the Quality of English LanguageIt need to be improved
Author Response
table 2 – we believe the location of median (IQR) is suitable.
Figures 2-4. The numbers are correct. The numbers are inconsistent between the different figures due to the nature of this being a landmark analysis. For instance, when looking at overall survival, any patient who died prior to day 180 would not be included in the landmark analysis (i.e. we cannot assess the survival time beyond day 180 if they died prior to day 180). However, for relapse-free survival, any patient who died or had a relapse before day 180 would not be included. Hence, the numbers should not match between OS and RFS.
Reviewer 2 Report (Previous Reviewer 3)
Comments and Suggestions for Authors
Although the author has made some revisions, the readability of the article still needs to be improved, and the current version of the article is not suitable for publication.
(1) In Fig2-Fig5, there are two identical pictures in each legend, making it difficult for readers to understand the pictures.
(2) The author generated a separate list of "conclusions", but it was not displayed in the revised manuscript.
Comments on the Quality of English LanguageModerate editing of English language required
Author Response
Comment 1: we apologize for the confusion. The duplicate figures were due to keeping track changes on, to allow the reviewers to see what changes were made. We have provide a ‘clean’ version for your review.
Comment 2:There is a list of conclusions in the manuscript attached.

This manuscript is a resubmission of an earlier submission. The following is a list of the peer review reports and author responses from that submission.
Round 1
Reviewer 1 Report
Comments and Suggestions for Authors
Little is known about the incidences and characteristics of multisystem immune-related adverse events (irAEs) associated with dual agent ipilimumab and nivolumab use. In this study, authors assess for rate, type, and timing of multisystem irAEs in 123 patients with melanoma or renal cell carcinoma with combination therapy of ipilimumab and nivolumab at our institution. In addition , authors also analyse the relation of multisystem irAE with progression-free survival (PFS) and overall survival (OS) indicating patients who experienced multisystem irAEs had significantly higher progression-free survival (PFS) and overall survival (OS) than those who experienced single or no irAE. Although the topic is interesting , there are some problems , especially some data is inconsistent in figure, table or text ,which can’t support the conclusion .
Major comments
1. Patients Information is not complete. Lack many major information about patients .For example. What is for therapy program for 1 cycle( or one does) 2, 3,4 of ipilimumab and nivolumab . Does of antibody and time between two cycles?
2. some description is unclear
1) What’s meaning of BRAF in Table1
2) The title for Figure 1. number of irAE(s) Per Person is misfit with content in figure 1 what is the percentage of different irAE(s) in all patients but not number of irAE(s) Per Person.
3)The percentage of different irAE(s) in table 2 is inconsistent with that in figure 1
4) 0,6,12,24----of horizontal ordinate in Figure 3,4,5,6 is months or weeks
2. Tables and figures are not standard and some data is inconsistent in figure ,table or description
1) The title of Figure should be at bottom of related figure not top or separation , which impact to read.
2) Some content in Tables lack
3) What’s meaning of All irAEs in Figure 2. Why does the rate of All irAEs is lower than 1 irAE patients and >=2 irAEs patients?
4) The percentage in curve in figure 3,4,5, 6 is inconsistent with data of bottom of Figure and description in text. So the Conclusion is Inaccurate
Comments on the Quality of English Language/
Reviewer 2 Report
Comments and Suggestions for Authors
In this retrospective study, Yuchen Li et al. used clinical patients’ samples and information to explore the correlation of irAEs and the use of dual immunotherapy agents. They provided thorough information and organized it in a presentable way. They also did further survival analysis to reveal the correlation. This study is of both novelty and clinical meaning. However, this paper cannot be accepted without the following modifications:
1. There was too much information included in the Introduction which makes the Introduction unnecessarily redundant. Paragraph 1 and 2 can be combined and simplified.
2. The structure of Results is confusing and chaotic, there were no clear orders of different sections of the Results. Figures, tables and manuscripts were not well organized as a whole.
3. The structure of Table 1 and 2 both needs to be improved. For example, why hospitalization rate was only shown in All Patients but not melanoma or RCC separately? The current tables are hard to read.
4. In Figure 1, the pie chart and histogram can be combined, and this piece of information is clearly exhibited in Table 2, it is unnecessary to show again.
5. Table 3 is very confusing to read. The head of Table 4 is black, and nothing can be read.
6. All the tables in Fig 3-6 can be deleted.
Comments on the Quality of English LanguageMinor editing of English language required.
Reviewer 3 Report
Comments and Suggestions for Authors
The study addresses an important gap in the knowledge regarding the incidences and characteristics of multisystem immune-related adverse events (irAEs) associated with the dual agent ipilimumab and nivolumab use. This is valuable information for the medical community and patients. There are several issues that need attention:
Authors should separate discussion and results.
It would be better if the author adds the SD value to the results.
Authors need to consider the effects of lifestyle habits such as smoking and drinking, as well as the effects of other medications.
Finally, the author should propose possible solutions or give feasible suggestions to avoid or reduce this risk.
Comments on the Quality of English LanguageMinor editing of English language required
